# Active Surrogate Estimators: An Active Learning Approach to Label–Efficient Model Evaluation

**Jannik Kossen**[1*]     **Sebastian Farquhar**[1,3]     **Yarin Gal**[1]     **Tom Rainforth**[2]

[1] OATML, Department of Computer Science, University of Oxford
[2] Department of Statistics, University of Oxford
[3] DeepMind

## Abstract

We propose *Active Surrogate Estimators (ASEs)*, a new method for label-efficient model evaluation. Evaluating model performance is a challenging and important problem when labels are expensive. ASEs address this active testing problem using a surrogate-based estimation approach that interpolates the errors of points with unknown labels, rather than forming a Monte Carlo estimator. ASEs actively learn the underlying surrogate, and we propose a novel acquisition strategy, *XWED*, that tailors this learning to the final estimation task. We find that ASEs offer greater label-efficiency than the current state-of-the-art when applied to challenging model evaluation problems for deep neural networks.

## 1  Introduction

Machine learning research routinely assumes access to large, labeled datasets for supervised learning. Practitioners often do not have that luxury. Instead, no labeled data might be available initially, and label acquisition often involves significant cost: it might require medical experts to analyze x-ray images, physicists to perform experiments in particle accelerators, or chemists to synthesize molecules. In these situations, we should carefully acquire only those labels that are most informative for the task at hand, maximizing performance while minimizing labeling cost [1, 62, 27, 43].

While a wide array of active learning methods [1, 62] have been developed to address the problem of *training* models when labels are expensive, much less attention has been given to the problem of label-efficient model *evaluation*. Such model evaluation is important, e.g. for performance analysis, model selection, or hyperparameter optimization. Indeed, Lowell et al. [45] argue that a lack of efficient approaches to model evaluation holds back adoption of active learning methods in practice.

Prior work in label-efficient model evaluation has focused on using importance sampling (IS) techniques to reduce the variance of Monte Carlo (MC) estimates [38, 59, 70]. In particular, recent approaches [38, 18] have iteratively refined the proposal using information from previous evaluations, producing adaptive IS (AIS) estimators with state-of-the-art performance.

Though these methods provide substantial label-efficiency gains over naive sampling, they have some clear weaknesses that suggest improvements should be possible. Firstly, the AIS weights assigned to samples depend on the acquisition proposal used *at that iteration*. Thus, early samples are often inappropriately weighted, because we are yet to learn an effective proposal. This can significantly increase the variance of the resulting estimator. Secondly, MC estimates can be inefficient at dealing with label noise: constraining the estimate to be a weighted sum of observed losses is limiting when the losses themselves are noisy. Finally, IS methods can be restrictive on the acquisition strategies used to choose points to label: acquisition must be stochastic with known probabilities.

---

*Correspondence to jannik.kossen@cs.ox.ac.uk.

36th Conference on Neural Information Processing Systems (NeurIPS 2022).

To address these limitations, we introduce **Active Surrogate Estimators (ASEs)**, a new approach to label-efficient model evaluation that is based on interpolation rather than MC estimation. Specifically, ASEs actively learn a surrogate model to directly predict losses at all observed *and unobserved* points in the pool. Here, the surrogate can take the form of any appropriate machine learning model, and we introduce a general framework for learning effective surrogates. The interpolation-based nature of ASEs leads to a variety of advantages over MC estimates: ASEs better accommodate noisy label observations, adapt quicker to new information, and provide complete flexibility in how new datapoints are acquired, allowing for careful customizations that are not appropriate in the AIS framework. To exploit this, we propose a novel acquisition strategy, **XWED**, that tailors the acquisition towards points that will be most beneficial to the final estimation task.

To summarize, our main contributions are: **(1)** We introduce Active Surrogate Estimators; a new method for label-efficient model evaluation based on actively learning a surrogate model (§3). **(2)** We introduce XWED, a novel acquisition strategy that targets a weighted disagreement that is highly relevant to our final surrogate-based prediction (§4). **(3)** We provide theoretical insights into ASEs' errors (§5). **(4)** We apply ASEs to the evaluation of deep neural networks trained for image classification, where we consistently outperform the current state-of-the-art (§7).

## 2  Label-Efficient Model Evaluation

We first describe the general setup of label-efficient model evaluation, also known as *active testing* [38]. We wish to evaluate the predictive model $f : \mathcal{X} \to \mathcal{Y}$, mapping inputs $x \in \mathcal{X}$ to labels $y \in \mathcal{Y}$. Here, model evaluation refers to estimating the expected loss of the model predictions, that is, the risk

$$r = \mathbb{E}\left[\mathcal{L}(f(X), Y)\right], \tag{1}$$

where we use capital letters for random variables, $\mathcal{L}$ is a loss function, and the expectation is over the true test distribution, which can be different to the distribution of the training data for $f$. We make no assumptions on $f$ or the way in which it has been trained. We only require that we can query its predictions at arbitrary $x$. Because we wish to estimate the risk of $f$, we keep $f$ fixed and it does not change during evaluation. Everything here applies to both classification and regression tasks.

Following existing approaches, we study active testing in the pool-based setting. This is the typical scenario experienced in practice [62]: samples $x_i$ can often be cheaply collected in the form of unlabeled data, but the density for the data generating process itself is typically unknown. Concretely, one assumes the existence of a pool of samples, with indices $\mathcal{P} = \{1, \dots, N\}$, drawn from the test distribution, and for which we know input features $x_i$ but not labels $y_i$, $i \in \mathcal{P}$.

If we knew the labels at all test pool indices $\mathcal{P}$, we could compute the empirical test risk $\hat{r} = 1/N \sum_{i \in \mathcal{P}} \mathcal{L}(f(x_i), y_i)$. When $N$ is large, this provides an accurate estimate of $r$. However, one often cannot afford to evaluate $\hat{r}$: the cost associated with label acquisition is too high to acquire labels for all points in $\mathcal{P}$. Instead, active testing methods must decide on a subset $\mathcal{O} \subseteq \mathcal{P}$ of points to label, where typically we have that $M = |\mathcal{O}| \ll |\mathcal{P}| = N$.

The simplest approach would be to sample indices uniformly at random. The resulting naive MC estimate, $\hat{r}_{\text{iid}} = 1/M \sum_{i \in \mathcal{O}} \mathcal{L}(f(x_i), y_i)$, represents the predominant practice of using randomly constructed test set instead of actively evaluating models. However, the variance of $\hat{r}_{\text{iid}}$ is often unacceptably high when $M$ is small. Methods for active testing seek to provide risk estimates with lower estimation error than $\hat{r}_{\text{iid}}$, without increasing the number of label acquisitions $M$.

Similar to active learning, it is advantageous in active testing to not select all points $\mathcal{O}$ upfront, but instead to select them adaptively by using information from previous evaluations. The mechanism by which points are chosen is known as an *acquisition strategy*. As with active learning, it is typical to assume that the cost of labeling is dominant, and all other costs incurred during model evaluation are typically ignored. Therefore, it can be worth (repeatedly) training surrogate models, despite the resulting expense, if this helps minimize the estimation error for a given label budget $M$.

### 2.1  Importance Sampling–Based Active Testing

Importance sampling (IS) [30, 51] is a classic strategy for forming MC estimates with lower variance than naive MC. IS samples points from a proposal $X \sim q$ instead of the original reference distribution, which we will denote $p$. Adaptive importance sampling (AIS) [32, 48, 5] improves IS by refining the proposal $q$ using information from previous iterations. However, traditional IS and AIS are only

applicable if we can evaluate the density of the reference distribution, at least up to normalization. This is not the case in many relevant active testing scenarios, e.g. when $x_i$ are high-dimensional images.

IS cannot be applied directly in the pool-based setting because of the need to sample *without* replacement. However, in the context of a training objective for active learning, Farquhar et al. [18] introduced LURE, which converts IS to this pool-based setting. Specifically, they compute

$$\hat{r}_{\text{LURE}} = \frac{1}{M} \sum_{m=1}^{M} v_m \, \mathcal{L}\left(f(x_{i_m}), y_{i_m}\right), \; v_m = 1 + \frac{N-M}{N-m}\left(\frac{1}{(N-m+1)q(i_m)} - 1\right), \quad (2)$$

where $q(i_m)$ is the probability of choosing to label sample $i_m$ at iteration $m$. Kossen et al. [38] showed that LURE can also be applied to active testing, for which the optimal proposal is

$$q^*(i_m) \propto \mathbb{E}[\mathcal{L}(f(X), Y)|X = x_{i_m}]. \quad (3)$$

In practice, $q^*$ is intractable as the distribution $Y|X = x_{i_m}$ is unknown. However, using ideas from AIS, one can adaptively improve $q$ with information gathered from previous labels. In particular, Kossen et al. [38] introduce strategies to directly approximate (3) (see §6). When $q$ correctly identifies regions in the input where the loss is large, $\hat{r}_{\text{LURE}}$ reduces variance compared to $\hat{r}_{\text{iid}}$.

Unlike standard AIS, adaptive LURE conveniently does not require knowledge of an explicit density $p(x)$—a pool of samples suffices. However, most techniques for reducing the variance of the weights in the AIS literature (see e.g. Bugallo et al. [5]) cannot currently be transferred over to adaptive LURE. In particular, we cannot employ methods like deterministic mixture AIS [50] to update historic weights using future proposal adaptations. This means that poorly-calibrated early weights can persist and significantly increase the variance of the estimator.

## 3 Active Surrogate Estimators

We now introduce Active Surrogate Estimators (ASEs), our novel method for label-efficient model evaluation. Following the setup in §2, we assume a pool of unlabeled samples from which we iteratively choose points to label. Unlike existing methods that rely on MC estimates, ASEs use an interpolation-based approach to estimate model risk: a surrogate model directly predicts losses for *all* points in the test pool. ASEs are further based around an adaptive and iterative procedure: we use an active learning approach to carefully select the labels used to train our surrogate. We first discuss the form that the ASE and surrogate take, and later cover our novel acquisition strategy in §4.

### 3.1 Deriving the ASE

The fundamental idea of ASEs is that of interpolation-based estimation: we wish to build a risk estimator of the form $\hat{r}_{\text{ASE}}(\phi) = 1/N \sum_{i \in \mathcal{P}} g(x_i; \phi)$, where $g(x_i; \phi)$ is a flexible surrogate model with parameters $\phi$ that approximates the conditional expected loss, $\mathbb{E}\left[\mathcal{L}(f(X), Y)|X = x\right]$.

Though algorithmically quite different, the motivation behind this interpolation-based approach is quite similar to that of variational inference (VI) [74]: rather than relying on a sample-based approximation for $\hat{r}$, we are converting our estimation into the optimization of a functional approximation. Though, as with VI, this comes at the cost of weaker theoretical asymptotic guarantees compared with MC estimation (see §5), it allows us to generalize more effectively when only a small number of labels are available. This trade-off is typically worthwhile in the small sample size regime of active testing.

To decide what form $g$ should take, we first note that the theoretically optimal $g$—in terms of mean squared error (MSE) for $r$—is simply the expected loss above, analogously to the optimal proposal for LURE. While this is itself obviously intractable, we can learn an effective $g$ by trying to approximate this optimal $g^*$. Though one could directly approximate this using a generic regressor, such as a neural network, this would disregard that we already know the loss function $\mathcal{L}$, can evaluate $f$, and know how both are combined to give the optimal $g^*$. In fact, the only unknown is the conditional expectation $Y|X = x$ for the test distribution.

We therefore introduce an auxiliary model $\pi(y|x; \phi)$ of only this conditional. Combining $\pi$, $f$, and $\mathcal{L}$, we construct $g$ as $g(x; \phi) = \mathbb{E}_{Y \sim \pi(\cdot|x;\phi)}\left[\mathcal{L}(f(x), Y)\right]$, where the only learnable components of $g$ are the parameters of $\pi$, $\phi$. With $\pi$ introduced, we can finally write the ASE as

$$\hat{r}_{\text{ASE}}(\phi) = \frac{1}{N} \sum_{i \in \mathcal{P}} \mathbb{E}_{Y \sim \pi(\cdot|x_i;\phi)}\left[\mathcal{L}(f(x_i), Y)\right], \quad (4)$$

where we emphasize that the expectation over $Y|X = x_i$ is now with respect to $\pi$ and not the unknown true label conditional. Note that the fact that the ASE includes an explicit model of the label noise is a decisive advantage over prior work (see §8). For classification, we can now directly evaluate the expectation over the labels as a weighted sum over the predictive probabilities $\sum_y \pi(y|x_i; \phi)\mathcal{L}(f(x_i), y)$. For regression, closed-form solutions of the expectation exist for some losses, and we can always compute unbiased and accurate MC estimates using cheap samples from $\pi$.

Assembling $g$ from $\mathcal{L}$, $f$, and $\pi$ will typically be advantageous to modeling $g$ directly: a direct model would need to learn not only about predictions of $f$, but also how they relate to the test distribution. In contrast, learning models $\pi(y|x; \phi)$ of conditional outcomes is standard procedure. This also makes it easy to include other prior information. For example, initializing $\pi$ on the *training* data can sometimes give the ASE a useful estimate of the risk before observing any test data at all. Lastly, the modular construction of $g$ allows us to quickly adapt the ASE to different models or loss functions.

Often, $f$, like $\pi$, will be a probabilistic model of outcomes. However, it will generally be a highly inappropriate choice for $\pi$, such that it is important to always learn a separate auxiliary model. This is because we are essentially using $\pi$ to predict the errors of $f$, and we should therefore aim to decorrelate the errors of $\pi$ from those of $f$ by training a separate $\pi$. Further, we may want to choose $\pi$ from a different model class than $f$, e.g. to allow for meaningful incorporation of uncertainties. Lastly, while $f$ is by construction fixed, we can adapt $\pi$ during evaluation to reflect our updated knowledge of the test label distribution, as we explain next.

## 3.2 Adaptive Refinement of ASEs

ASEs are adaptive: they iteratively improve their estimate by updating the surrogate as new labels are acquired. A crucial component to them doing this effectively is that they actively select the labels to acquire at each iteration using an acquisition strategy. As such, they can be thought of as performing estimation via active learning of the auxiliary model $\pi$, noting that the estimate they produce is, in turn, fully defined by this auxiliary model and the pool set.

---

**Algorithm 1** Adaptive Refinement of ASEs

**Inputs:** Test pool $\{x_i\}_{i \in \mathcal{P}}$, acquisition strategy $a$, target model $f$, initial auxiliary model $\pi_0$
1: Initialize $\mathcal{O} = \emptyset$ and $\mathcal{U} = \mathcal{P}$
2: **for** $m = 1$ to $M$ **do**
3:      Select $i_m = a(\pi_{m-1}, f, \{x_i\}_{i \in \mathcal{P}}, \{y_i\}_{i \in \mathcal{O}})$
4:      Acquire label $y_{i_m} \sim Y \mid x_{i_m}$
5:      Update $\mathcal{O} \leftarrow \mathcal{O} \cup i_m$ and $\mathcal{U} \leftarrow \mathcal{U} \setminus i_m$
6:      Update auxiliary model $\pi_m$ (e.g. by retraining)
7: **end for**
8: Return $\hat{r}_{\text{ASE}}$ as per (4) with $\pi = \pi_M$

---

To define this adaptive refinement process more formally, we denote with $\mathcal{U}$ and $\mathcal{O}$ the sets of unobserved and observed points in the test pool. These are initialized as $\mathcal{U} = \mathcal{P}$ and $\mathcal{O} = \emptyset$. At each subsequent iteration $m$ of our adaptive refinement process, an acquisition strategy $a$ selects an index $i_m \in \mathcal{U}$ of an unlabeled datapoint, for which we then acquire a label $y_{i_m}$. We remove $i_m$ from $\mathcal{U}$, and add it to $\mathcal{O}$. Then, we obtain an improved ASE by retraining $\pi$, e.g. on the total observed test data $\{(x_i, y_i) : i \in \mathcal{O}\}$; this retraining does not necessarily need to be performed at every iteration if doing so would be computationally infeasible. A summary of this approach is given in Algorithm 1.

In §E, we show that the computational complexity of ASEs is given by $\mathcal{O}\left(M \cdot (N + |\mathcal{D}_{\text{train}}|)\right)$, assuming an initial training set of size $|\mathcal{D}_{\text{train}}|$ for the surrogate. This is identical to the computational complexity of LURE-based active testing but more expensive than a naive MC estimate.

In ASEs, our acquisition strategy takes the current auxiliary model $\pi_m$, the original model $f$, the pool of samples, and all previous evaluations as input, i.e. we have $i_m = a(\pi_{m-1}, f, \{x_i\}_{i \in \mathcal{P}}, \{y_i\}_{i \in \mathcal{O}})$. A good acquisition strategy selects points such that the expected ASE error after $M$ iterations is minimal [28]. While ASEs can be used with arbitrary acquisition strategies, we next introduce a novel acquisition strategy that we specifically design for use with ASEs.

## 4 The XWED Acquisition Function

It is important we tailor our acquisition strategy to the problem of active surrogate estimation, rather than apply more generic active learning approaches. We need to acquire labels such that we produce the best ASE estimate given a limited labeling budget. To do this, we propose a novel acquisition function called the **Ex**pected **We**ighted **D**isagreement, or *XWED* for short, that is designed specifically to target improvements of the ASE estimate itself.

In line with acquisition strategies in active learning, we now require that $\pi$ takes the form $\pi(y|x;\phi) = \mathbb{E}_{\Theta\sim\pi(\cdot;\phi)}[\pi(y|x,\Theta;\phi)]$, where $\Theta$ represents some model parameters we wish to learn about. In a BNN, $\Theta$ is the weights and biases of the network, while $\phi$ corresponds to the parameters of the posterior approximation given by previous observations (such that it contains all information from previous data). To avoid clutter, we omit $\phi$ in the following presentation of acquisition functions.

For active learning, BALD [27] is an established acquisition strategies that acquires points with high epistemic uncertainty [13, 34]. For a predictive model $\pi$ as above, BALD can be calculated as the disagreement between the total predictive entropy and the expected entropies over posterior draws,

$$\text{BALD}(x) = \mathbb{E}_{Y\sim\pi(\cdot|x)}\left[-\log\pi(Y|x)\right] - \mathbb{E}_{\Theta\sim\pi(\cdot)}\left[\mathbb{E}_{Y\sim\pi(\cdot|x,\Theta)}\left[-\log\pi(Y|x,\Theta)\right]\right]. \quad (5)$$

However, BALD, and other acquisition functions from active learning, are ill-suited to ASEs because they do not consider the effect the uncertainties in $\pi$ have on the ASE estimate. In particular, ASEs depend on the loss $\mathcal{L}(y, f(x))$ of the original model $f$ at the pool samples, but neither the form of the loss function $\mathcal{L}$ nor the original model $f$ are considered in BALD acquisition.

Acquisition for ASEs should instead target points that are both informative about $\Theta$ *and* which we expect will contribute significantly to our final risk estimate. The basis of XWED is therefore to *weight* the disagreement terms for particular inputs $x$ using their corresponding loss $\mathcal{L}(Y, f(x))$:

$$\text{XWED}(x) = \mathbb{E}_{Y\sim\pi(\cdot|x)}\left[-\mathcal{L}(Y, f(x))\log\pi(Y|x)\right] \quad (6)$$
$$- \mathbb{E}_{\Theta\sim\pi(\cdot)}\left[\mathbb{E}_{Y\sim\pi(\cdot|x,\Theta)}\left[-\mathcal{L}(Y, f(x))\log\pi(Y|x,\Theta)\right]\right].$$

With XWED, we acquire points with large epistemic uncertainty [13, 34] in the auxiliary $\pi$ about the outcomes *only if they are relevant to the estimate* $\hat{r}_{\text{ASE}}$. To convert this acquisition function to an acquisition strategy, we simply choose the point for which the XWED score is maximal: $a(\pi_{m-1}, f, \{x_i\}_{i\in\mathcal{P}}, \{y_i\}_{i\in\mathcal{O}}) = \arg\max_{i\in\mathcal{U}}\text{XWED}(x_i)$. As we show in §7, acquisition from XWED allows for significant improvements compared to BALD and other strategies.

XWED is compatible with a variety of model classes for $\pi$, such as Bayesian Neural Networks (BNNs) [46, 21] and deep ensembles [41]. In §D.2, we give a worked through example for the practical computation of XWED when the main model $f$ is a neural network and the surrogate $\pi$ a deep ensemble, both trained for classification. We emphasize that XWED is an acquisition strategy explicitly designed for active testing; it is not even defined in a traditional active learning context, where the distinction between a fixed model $f$ and the adaptive auxiliary model $\pi$ does not exist.

## 5   Theoretical Analysis of the ASE Error

We next provide a theoretical analysis of ASEs to understand individual contributions to their error. Specifically, we will break down the errors of ASEs into $(I)$ error from using a finite evaluation pool and $(II)$ error from imperfection of the surrogate, further breaking the latter down as $(IIA)$ error from any limited expressivity in our surrogate class and $(IIB)$ error from imperfect training.

To start, let $\Phi$ be the final parameters obtained from the stochastic training procedure of the surrogate $g$, such that our learned surrogate is $g(\cdot; \Phi)$. Here, $\Phi$ is a random variable with the stochasticity originating from (i) sampling the test pool itself, (ii) any stochasticity in the acquisition strategy, (iii) randomness in the acquired labels, and (iv) stochasticity in the optimization process of the surrogate.

Next, let $X_0 \sim p(x)$ be a hypothetical additional input sample that is independent from the test pool. We can use this to define $G(\phi) = \mathbb{E}[g(X_0; \phi)]$ as the true expectation with respect to the input distribution of a surrogate with parameters $\phi$. In turn, we can define $\phi^* = \arg\min_\phi |G(\phi) - r|$ as the parameters which give the best approximation of the true expected loss $\mathbb{E}[\mathcal{L}(f(X), Y)|X = x]$. Note here that $\phi^*$ is not random as it depends only on the fixed function class of $g$ and the fixed true risk $r$.

Denoting our test pool as $P = \{X_1, \ldots, X_N\}$, where $X_i \sim p(x)$ are independently sampled, we can write $\hat{r}_{\text{ASE}}(\Phi, P)$ to make the dependency of the ASE estimator (4) on $P$ and $\Phi$ explicit. By using the triangle inequality, we can then break down the error $\|\hat{r}_{\text{ASE}}(\Phi, P) - r\|$ to the true model risk $r$ as

$$\|\hat{r}_{\text{ASE}}(\Phi, P) - r\| = \|\hat{r}_{\text{ASE}}(\Phi, P) - G(\Phi) + G(\Phi) - G(\phi^*) + G(\phi^*) - r\|$$
$$\leq \|\hat{r}_{\text{ASE}}(\Phi, P) - G(\Phi)\| + \|G(\Phi) - r\|$$
$$\leq \|\hat{r}_{\text{ASE}}(\Phi, P) - G(\Phi)\| + |G(\phi^*) - r| + \|G(\Phi) - G(\phi^*)\|, \quad (7)$$

where $\|\cdot\|$ is any valid norm, and we note that $G(\phi)$ is independent of $X_0$ and $G(\phi^*)$ is deterministic.

We now consider each of the terms in (7):

$(I) = \|\hat{r}_{\text{ASE}}(\Phi, P) - G(\Phi)\|$ is the error that originates from only evaluating the surrogate on a finite set of pool of points. As per standard results for MC estimation, this will generally decrease as $\mathcal{O}(1/\sqrt{N})$, such that it will be small for large test pools. Note here that $\hat{r}_{\text{ASE}}(\Phi, P)$ is typically not an unbiased estimator for $G(\Phi)$ as the set of input points available in the pool can influence the distribution of $\Phi$. However, this effect will typically be very small and the bias will diminish as $N$ increases. We can also, if desired, eliminate it completely by evaluating $g$ over a separate test pool than that used during the acquisition of test labels and updating of $g$. This variation on the standard ASE estimator can also be useful when the pool is impractically large for performing label acquisition, as we can evaluate the final learned $g$ on a larger pool than used during its training.

$(II) = \|G(\Phi) - r\|$ represents the error from the imperfection of the surrogate. For the squared norm, it equals $\mathbb{E}_{\Phi}[(\mathbb{E}_{X_0}[g(X_0; \Phi)] - r)^2]$, such that it is the expected squared difference (over surrogate parameters) between the surrogate's expectation over inputs and the true risk. It can itself be further broken down into terms $(IIA)$ and $(IIB)$ as described below. Note that this term demonstrates the critical dependency of ASE's performance on the regressional performance of the surrogate itself.

$(IIA) = |G(\phi^*) - r|$ gives the error from limits in the choice of the surrogate function class. It measures the difference between $g^*$ and $g(\cdot; \phi^*)$. Whenever our surrogate function class contains the true expected loss, this term will be exactly zero. Even when this is not the case, it should generally be negligible for sensible choices of the surrogate function class. For example, if $\pi$ is a deep neural network, the limiting factor will almost always be the training error from using finite data (i.e. term $(IIB)$ below), rather than the theoretical expressivity of the network.

$(IIB) = \|G(\Phi) - G(\phi^*)\|$ can be thought of as the error originating from the training of the surrogate: it is the norm of the difference between the true expectation of the surrogate we have learned and the true expectation of the best possible surrogate we could have learned.

## 5.1 Discussion and Empirical Analysis

In practice, $(IIB)$ will typically dominate the ASE error. If term $(I)$ is significant, the pool size is a limiting factor, which can only be the case when our ASE estimation has been successful (assuming $M \ll N$), as its implies we have a highly accurate estimate of the full empirical test risk $\hat{r}$. If term $(IIA)$ is significant, this means we are using an inappropriately weak model class for our surrogate.

To check these assertions hold in practice, we empirically estimate $(I)$ and $(IIA)$ for the experiments in §7.1 at $M = 1$: we find that they respectively make up only $0.00013\%$ and $0.031\%$ of the full error; thus $(IIB)$ is completely dominant here.

Further characterizing the dominating term $(IIB)$ in the fully general setting is unfortunately not feasible: it equates to characterizing the error of a regression, which necessarily can only be done by considering a particular class of surrogate. However, for a particular choice of surrogate, existing results from learning theory could be applied to further categorize its convergence. Here work on the convergence of models in active *learning* [23, 24, 58, 7, 54] will be particularly appropriate, noting that these all make specific assumptions about the model and acquisition strategy. More specifically, for Gaussian process surrogates and acquisition strategies from Bayesian quadrature, the results of Kanagawa & Hennig [31] could be applied. On the other hand, for surrogates based on deep learning architectures, one could use results from the convergence of stochastic gradient descent methods [47], which typically give rates of $\mathcal{O}(1/\sqrt{M})$ under appropriate assumptions. These will require unbiased gradients in the optimization of the objective function, which we can achieve for ASEs with stochastic acquisition and the de-biasing scheme of Farquhar et al. [18].

In summary, for large pool sizes and appropriate surrogate classes, $(IIB)$ will be the dominant contributor to the error, noting that the inequality in (7) becomes tight as one of the errors dominates. As such, the convergence of ASEs essentially reduces to the convergence of the surrogate, and thus, by extension, the convergence of the auxiliary model. Moreover, if we can ensure that this auxiliary model convergences to the true conditional output distribution as the number of label acquisitions $M$ increases (this itself implies that our surrogate function class is sufficient powerful and so $(IIA) = 0$), we can thus ensure that the ASE estimator itself converges, noting that $M \to \infty$ also predicates $N \to \infty$ (as $N > M$ by construction), such that $(I) \to 0$.

# 6 Related Work

**Active Learning.** Active learning (AL) [1, 62, 27, 21, 61] allows for the training of supervised models in a label-efficient manner, and it is a particular instance of optimal experimental design (OED) [44, 6, 60, 20]. OED formalizes the idea of targeted acquisition to maximize the information gained with respect to a target objective. Relatedly, transductive AL [71, 68, 36] performs AL of model predictions with respect to a known set of points. Filstroff et al. [19] propose AL for decision making.

**Active Testing.** Sawade et al. [59] explore IS for active risk estimation. They rely on vanilla IS which does not respect the pool-based setting [38]. Yilmaz et al. [70] have recently proposed active testing based on Poisson sampling. Both [70] and [59] outperform simple baselines; however, they use *non-adaptive* acquisitions that cannot adjust to test data.

Of particular relevance is the current state-of-the-art approach by Kossen et al. [38]. Building on the LURE estimator, they propose an AIS-style estimate of the model test risk (§2.1). To approximate the optimal acquisition (3), they, like us, introduce an auxiliary model $\pi(y|x; \phi)$ that is (usually) adapted as test labels are acquired. Given restrictions of the LURE estimator, they always sample from

$$q(i_m) \propto \mathbb{E}_{Y \sim \pi(\cdot|x_i; \phi)} [\mathcal{L}(f(x_{i_m}), Y)] \tag{8}$$

for label acquisition and do not actively improve the proposal, e.g. through active learning.

[2, 33, 40, 29] explore *stratification* for efficient evaluation of classifiers: a simple, non-adaptive criterion is used to divide the test pool into strata, from which labels are selected uniformly at random. These approaches can be improved by applying the method considered here within the strata.

**Surrogates.** The use of surrogate models to replace expensive black-box functions is pervasive throughout much of the sciences [53, 37, 10]. Bayesian optimisation (BO) [63, 4] and Bayesian quadrature (BQ) [49, 56, 3] are prominent examples of surrogate-based approaches that actively select samples. BO iteratively finds the maximum of an unknown function, using a surrogate model and ideas similar to active learning to select sample locations. Unlike active learning, BO does not focus on a globally accurate surrogate. BQ obtains a Bayesian posterior for the integration of a Gaussian process (GP) [55] surrogate against a known data density $p(x)$. Despite biased estimates, BQ can drastically outperform MC in low dimensions. However, BQ does not scale well to high dimensions, where, often, GP surrogates are not appropriate and explicit densities $p(x)$ are unknown and hard to approximate.

**Model Evaluation without Labels.** Similar to our experiments in §7.3, [22, 57, 9, 11, 12, 64, 65] study the evaluation of model test performance without test labels or even without test sets. However, unlike ASEs, these works are based around performing meta analysis and require access to multiple related datasets (for which test labels are available). They rely on regressions from dataset/model summaries to performance scores and cannot be used in the setting considered in this paper, where only one dataset is available. Similarly, meta-learning for hyperparameter optimization, e.g. [15, 67, 16, 72, 26], interpolates test performance across a variety of datasets *and* model hyperparameters.

# 7 Experimental Results

We next study the performance of ASEs for active testing in comparison to relevant baselines. Concretely, we compare to naive MC and the current state-of-the-art LURE-based active testing approach by Kossen et al. [38]. As a general and common application, we consider the important problem of actively evaluating deep neural networks trained on high-dimensional image datasets.

We give full details on the experiments in the appendix. In particular, §B and §C contain additional results and figures, and §D gives further details on the computation of XWED and the baselines.

## 7.1 Evaluation with Distribution Shift

We first explore model evaluation in a challenging distribution shift scenario. Concretely, we are given a fixed model trained on 2000 digits of MNIST [42], where all sevens have been removed from the training set. We then estimate the model risk on a test set with all classes present in their normal proportions. The test set has size $N = 60000$ but we can only afford to acquire labels for a small subset. Such distribution shift settings are relevant in practice *and* challenging for active testing: both ASE- and LURE-based active testing needs to approximate the expected loss, and this is particularly hard under distribution shift, where the initial $\pi$ has yet to learn about the test distribution.

Following [38], we use radial Bayesian Neural Networks [17] for $f$ and $\pi$, with $\pi$ initialized on the training set and regularly retrained during evaluation. For LURE, we acquire labels by sampling from (8). For the ASE, we use the XWED acquisition strategy (6). Both ASE and LURE use the same auxiliary model class $\pi$, but arrive at different network weights, $\phi_{ASE}$ and $\phi_{LURE}$, because their distinct acquisition strategies lead to different test observations used for the re-training of $\pi$.

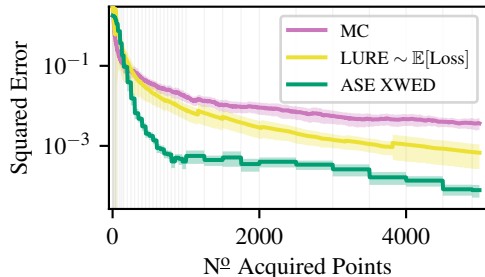

Figure 1: ASE significantly improves over LURE and MC in terms of squared error for the distribution shift experiment. Shown are mean errors and the shading is two std. errors over 100 runs. We retrain $\pi$ at steps marked with grey lines.

**Main Result.** Figure 1 shows the MSE for ASE and baselines when estimating the unknown test pool risk. Note here that we are careful to use the best possible setup of the LURE baseline to construct a challenging benchmark, as shown in the next section. We see that ASE improves on all baselines when used with the XWED acquisition strategy. Further, LURE can struggle with variance, with errors of individual runs occasionally spiking. This occurs when there is a significant mismatch between observed loss and proposal, e.g. when an acquired point is ambiguous or mislabeled. This does not happen for ASEs, as we do not weight true observations against predictions in the estimate.

## 7.2 Acquisition Strategies for ASE and LURE

We next perform an ablation to investigate different acquisition strategies besides XWED in the distribution shift scenario: We select points with maximal BALD score, Eq. (5). Further, we acquire points by sampling from the expected loss; this is the exact same acquisition strategy used for LURE-based active testing, cf. (8). See §D.1 for further details on how we compute these baselines.

Figure 2 visualizes how often the acquisition functions acquire the missing class as testing progresses, and Figure 3 (a) shows that ASE is able to outperform naive MC for *all* acquisition strategies, but there are noticeable variations in their performance. The XWED acquisition, that we have proposed specifically with ASEs in mind, provides the best performance. BALD acquisition does not target those points that lead to the largest improvements in the estimate, which leads to a much slower decrease in ASE error early on, though it does catch up with XWED later. Expected loss acquisition—which is optimal for LURE—will sample the missing class (where loss is high) almost exclusively, which

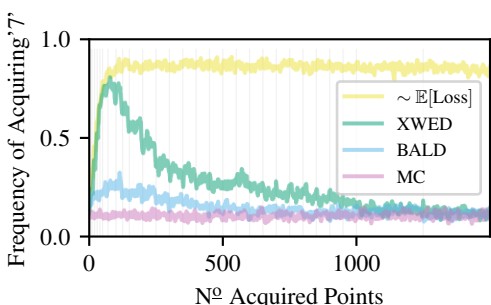

Figure 2: Proportion of acquired points which have class '7'. Results averaged over 100 runs.

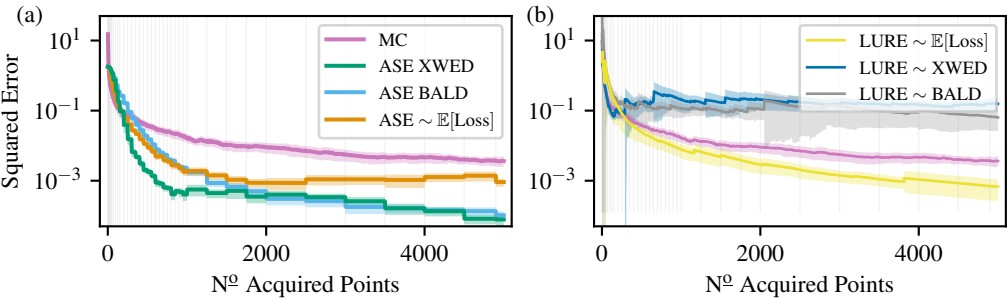

Figure 3: Different acquisition strategies for ASEs and LURE. (a) While XWED performs best, ASE outperforms MC for all investigated acquisition functions. (b) LURE suffers high variance for all acquisition strategies except expected loss. Shown are mean errors, shading is two std. errors over 100 runs. We retrain $\pi$ at steps marked with grey lines. The '$\sim$'-symbol indicates stochastic acquisition.

initially works well but later leads to $\pi$ failing to improve on the other classes. In contrast, XWED focuses on the missing class early on but—once the loss is well characterized for the missing class—explores other areas as well. This behavior is preferable over the baselines, and we continue this discussion in §B.1.

In Figure 3 (b), we explore how LURE behaves under different acquisition strategies, such as sampling from XWED or BALD scores. We find that LURE is noticeably more sensitive to the acquisition strategy than ASE: both XWED and BALD acquisition lead to high-variance estimates that no longer outperform the naive MC baseline. It seems one cannot expect acquisition strategies other than expected loss to perform well with LURE. In contrast, ASEs can directly apply any acquisition scheme without introducing such sensitivity because the acquisition does not affect which points are included in the estimate or how they are weighted.

We provide additional results in the appendix: §B.2 shows that adaptive improvement of the acquisition function is necessary for label-efficient active testing in the distribution shift scenario, and, in §B.3, we study additional choices for ASE acquisition strategies (which do not outperform XWED).

### 7.3 Evaluation of CNNs for Classification

Next, we study active testing of ResNets on more complex image classification datasets. Concretely, we study the efficient evaluation of a Resnet-18 [25] on Fashion-MNIST [69] and CIFAR-10 [39], and of a WideResNet [73] on CIFAR-100. In each case, we train the model to be evaluated, $f$, on a training set containing $40\,000$ points, and then use an evaluation pool of size $N = 2000$. As a ground truth, we compare to the empirical risk on a held out set of $20\,000$ points. For $\pi$, we use deep ensembles [41] of the original ResNet models. This experiment design is also used by [38].

In this experiment, there is no distribution shift between the training data for $f$ and the data on which we evaluate it. At the same time, the training set is large and, therefore, a lot of prior information is available to train the initial $\pi$. Further, the number of test acquisitions is comparatively small ($M = 50$) and unlikely to affect $\pi$ meaningfully, given the models have already seen $40\,000$ points from the training data. Indeed, we find that updating $\pi$ with this additional test data makes no noticeable difference to performance. To minimize computational costs, we therefore report results with $\pi$ fixed throughout this experiment, such that, here, $\pi$ is equal for ASE and LURE. A constant $\pi$ leads to a constant LURE proposal (making the method more akin to IS than AIS) and constant prediction for ASE.

Figure 4 shows that ASEs perform effectively in this setting, significantly outperforming LURE and naive MC for all datasets. We also display the error from the finite pool size, by comparing the empirical risk of all $N$ unobserved test points against our additional ground truth set and see that ASE consistently performs similarly to this pool limit. This suggests that the pool size $N$ may be the limit of the performance for ASEs here rather than the quality of the surrogate.

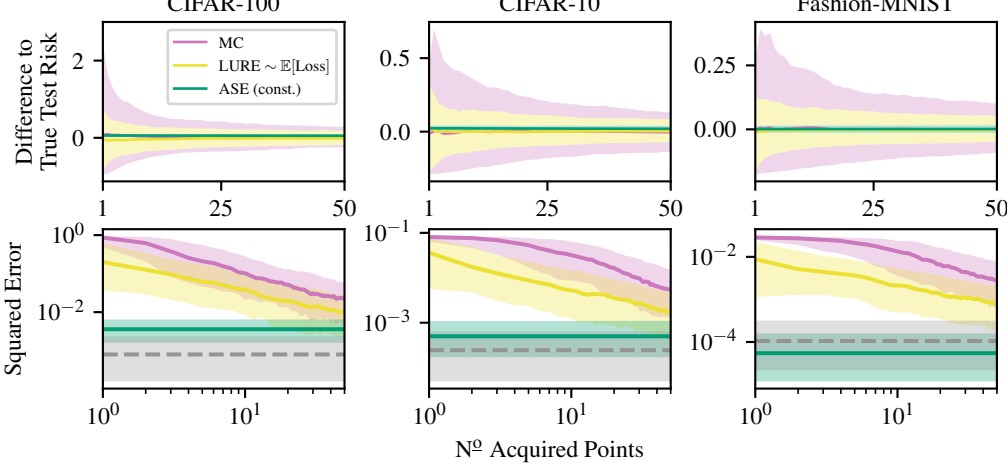

Figure 4: Active Testing of ResNets on CIFAR-100, CIFAR-10, and FashionMNIST without retraining the auxiliary model $\pi$. Despite not observing any test labels, ASE again improves upon LURE. We display medians over 1000 random test sets and 10/90 % (top) and 25/75 % (bottom) quantiles (top: too small to be visible for ASE). In grey, we display a lower bound on the error from the finite pool.

This experiment produces a rather unusual result: as $\pi$ is fixed throughout, the ASE actually does not make use of *any* information from the test labels! Its accurate estimates of the test risk are entirely due to information from the training data. Note that LURE here has access to the same $\pi$ as ASE in addition to up to 50 test labels. Nevertheless, the surrogate prediction of the ASE at all pool points leads to much lower error than a LURE estimate. This 'zero-shot' setting opens up interesting avenues for model evaluation when there are no test labels available at all.

However, in practice, one may not know if the test data follow the same distribution as the training data. Figure B.1 (a) shows that both LURE and ASE perform poorly when not retraining $\pi$ in the distribution shift setting for any labeling budget $M$. We therefore generally suggest regular adaptation of $\pi$ for both ASE- and LURE-based active testing approaches. And even without distribution shift, the information from test labels will always eventually become relevant as $M$ increases.

ASEs are compatible with arbitrary loss functions, cf. (4). For example, for 0-1-Loss, ASEs can be used to estimate the accuracy of the main model. In §B.4, we demonstrate that ASEs continue to outperform all other baselines for the task of accuracy estimation.

In our theoretical analysis in §5, we argue that, in most practical settings, the error originating from the surrogate will dominate the ASE error. In §B.5, we reduce the size of the training set to $10\,000$. This makes the problem more challenging for ASEs because it reduces the quality of the surrogate model. Nevertheless, we find that ASEs maintain their significant advantage over the baselines.

## 8  Discussion

In the previous section, we have seen ASEs outperform MC-based alternatives in practice. Here, we discuss advantages of ASEs that are important to their practical success.

**Actively Learning Estimators.** ASEs are flexible about which active learning strategy can be used. In contrast, AIS-based strategies must balance competing goals: the acquisition strategy must both find informative data to improve the proposal in future iterations and provide importance-sampling weights that reduce the estimator variance. For AIS, acquiring the most informative data with high probability can make the estimator *worse* because the resulting acquisition probabilities are a bad approximation for the optimal $q^*$. Prior work using LURE does not address this trade-off and always acquires with the best approximation of $q^*$ available. ASE acquisitions can also be deterministic, while AIS requires full-support stochastic acquisition to avoid bias.

**Surrogate Updates.** For AIS, improving the proposal distribution only improves future acquisitions, but does not retrospectively improve weights attached to earlier observations. Bad initial acquisition weights have a lasting influence on the variance of AIS. In contrast, ASE makes better retrospective use of information: the iterative refinement of $g$ leads to updated predictions at all pool points.

**Noisy Labels.** ASEs have a further advantage when label distributions are *noisy*, i.e. observations $y$ are stochastic even for the true label conditional $p(y|x)$. This is common in practice, because of ambiguous inputs near class-boundaries, mislabeled examples, or measurement error. ASEs directly model the noisy outcome conditionals using $\pi$. If $\pi$ is accurate, the ASE error does not increase due to noisy labels, and the ASE can interpolate beyond the noise. In contrast, AIS suffers badly: noisy observations form a direct part of the estimate, leading to increased variance; while the acquisition (8) models the noise, this only affects weights, and noisy loss observations directly enter the estimate (2). Chen & Choe [8] provide a detailed analysis of how stochastic observations affect IS estimators negatively.

## 9  Conclusions

We have introduced Active Surrogate Estimators (ASEs), a surrogate-based method for label-efficient model evaluation. The surrogate is actively learned, and we introduced the XWED acquisition function to directly account for the risk prediction task. Our experiments show that ASEs and XWED acquisition give state-of-the-art performance for efficient evaluation of deep image classifiers.

## Acknowledgments and Disclosure of Funding

We thank Andreas Kirsch and the anonymous reviewers for helpful feedback and interesting discussions that have led to numerous improvements of the paper. We acknowledge funding from the New College Yeotown Scholarship (JK) and the Oxford CDT in Cyber Security (SF).

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
