# Appendix

# Active Surrogate Estimators: An Active Learning Approach to Label–Efficient Model Evaluation

## A   Code

We release the full code for reproducing the experiments at https://github.com/jlko/active-surrogate-estimators.

## B   Additional Results

### B.1   Comparison of Acquisition Behavior

We next investigate how XWED outperforms all competing acquisition strategies in the experiments of §7.1. As a reminder, in the distribution shift setup the model $f$ is not exposed to any samples of class '7' during training, while the test set contains '7's in their normal proportion. Hence, predictions at samples with class '7' should contribute significantly to the risk of $f$ on the test distribution. However, the overall number of training points for $f$ is relatively small such that contributions from other classes to the risk are not negligible.

In Figure 2, we examine the probability of acquiring a '7' as a function of the number of acquired points for XWED, BALD, expected loss acquisition, and a uniformly at random sampling strategy (MC). We see that XWED initially focuses on 7s but then diversifies. The baselines always prioritize 7s or never do. The XWED behavior is preferable: we are initially unsure about the loss of these points, but once the loss is well characterized for the 7s we should explore other areas as well. Note also that this highlights the benefit of ASEs over LURE: LURE, which requires expected loss acquisition, inefficiently needs to keep giving preferences to 7s even once the loss for these is well characterized.

### B.2   Constant $\pi$ Fails for Distribution Shift.

We investigate if retraining of the auxiliary model $\pi$ is necessary to achieve competitive active testing performance in the distribution shift scenario. Here, we train $\pi$ once on the original training data and then keep it constant throughout testing, not updating $\pi$ with any information from the test data. ASE and LURE thus use the exact same $\pi$. When $\pi$ is constant, the ASE *estimate* and the LURE *proposal* are constant. For LURE, we acquire by sampling from (8) as usual. Figure B.1 (a) shows that, for both ASE- and LURE-based active testing, naive MC cannot be outperformed when $\pi$ is fixed. For this distribution shift scenario, it therefore seems that a fixed $\pi$ is neither informative enough as a proposal for LURE, nor does it allow good ASE estimates. This justifies our focus on *adaptive* approaches for active testing.

### B.3   Additional Acquisition Strategies for ASEs

For completeness, we here also investigate the performance for ASE when performing *stochastic* acquisition from the XWED and BALD scores. These are the exact acquisition strategies for which LURE suffered high variance in Figure 3. In Figure B.1 (b), we observe that ASE continues to perform well for both these acquisition strategies, although they do not outperform our default strategy that uses deterministic acquisition of the maximal XWED scores.

### B.4   Estimating Accuracy with ASEs

ASEs are compatible with arbitrary loss functions $\mathcal{L}$. For 0-1-Loss, $\mathcal{L}(f(x), y) = \mathbf{1}[f(x) = y]$, ASEs can be used to estimate the accuracy of the main model. We perform these experiments in the setup of §7.3. Figure B.2 demonstrates that ASEs continue to outperform all other baselines for the task of accuracy estimation. The accuracies of the main model in these experiments are $(73.31 \pm 0.93)\%$ for CIFAR-100, $(91.07 \pm 0.62)\%$ for CIFAR-10, and $(93.10 \pm 0.53)\%$ for Fashion-MNIST.

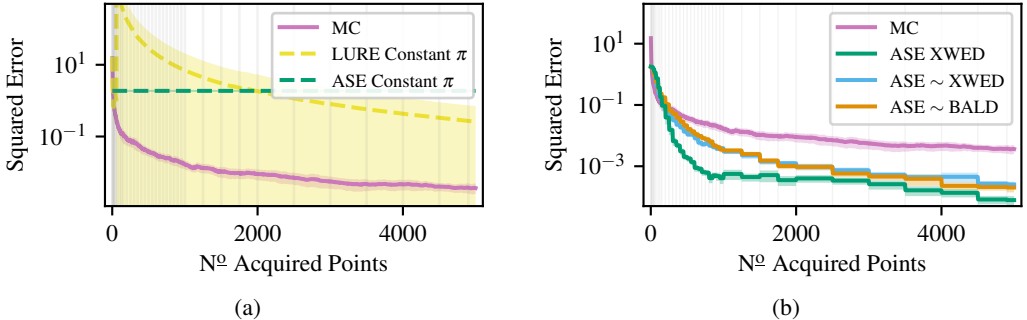

(a)                                                                (b)

Figure B.1: (a) If $\pi$ is constant, both ASE and LURE do not outperform naive MC in the distribution shift scenario. This result highlights the importance of the *adaptive* nature of both ASE- and LURE-based active testing. (b) Additional acquisition functions for ASE in the distribution shift scenario: while deterministic acquisition from XWED performs best, ASEs also give good performance when acquiring *stochastically* from XWED or BALD scores. This is unlike LURE, which is highly sensitive to the correct choice of acquisition function. We display means over 100 runs and shading indicates two standard errors (too small to be visible for some of the lines). The vertical grey lines mark iterations at which we retrain the secondary model.

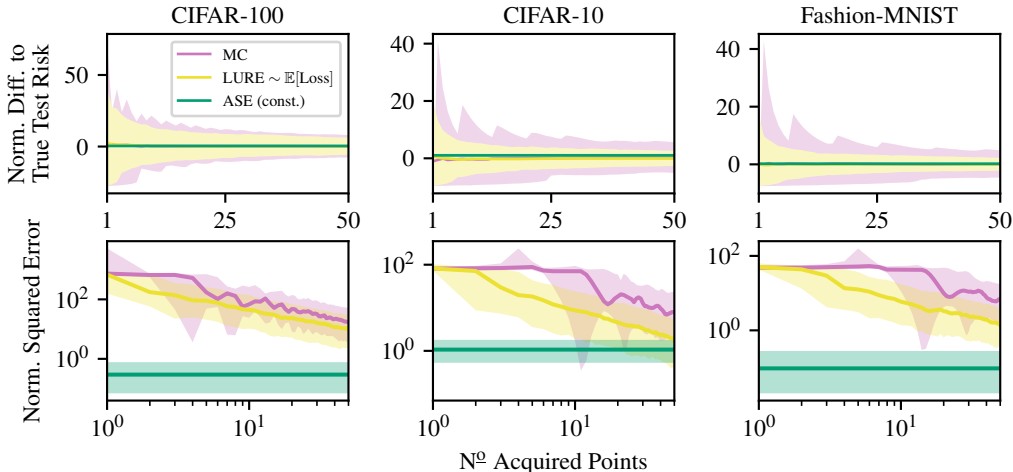

Figure B.2: Variant of the experiments of §7.3 where we estimate the *accuracy* of the main model. We display medians over 1000 random test sets and 10/90 % (top) and 25/75 % (bottom) quantiles (top: too small to be visible for ASE).

## B.5 Reduced Training Set Size for Experiments of §7.3

We here investigate a variation of the experiments in §7.3: reducing the size of the training set to 10 000 samples. This reduces the quality of the surrogate model and thus makes the problem more challenging for ASEs. Despite this, Figure B.3 demonstrates that ASEs continue to outperform all baselines. Again, note that the ASE does not use any test labels here and is therefore constant. Retraining of the ASE surrogate on the newly observed test data (with labels) will be necessary for ASEs to be able to continue to improve over the baselines at larger number of acquired test labels.

## C   Additional Figures

In the main body of the paper, we display both median and mean squared errors across the different figures to provide a variety of perspectives on ASEs. Here, we provide complimentary versions: for figures that show median behavior in the main paper, we here show means, and vice versa.

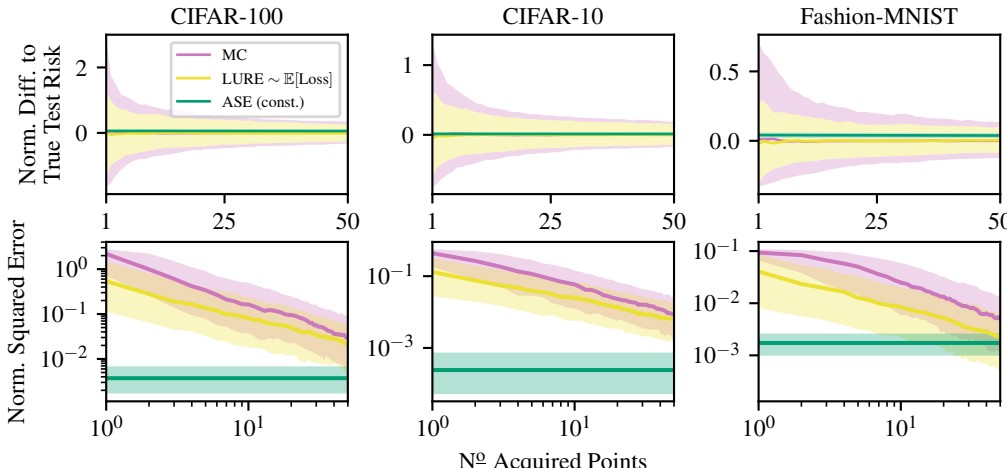

Figure B.3: Variant of the experiments of §7.3 where the training set size is reduced to 10 000 samples. We display medians over 1000 random test sets and 10/90 % (top) and 25/75 % (bottom) quantiles (top: too small to be visible for ASE).

See Figure C.1, Figure C.2, Figure C.3, and Figure C.4 for complimentary versions of the plots from the experimental evaluations in §7 and §B.

For ASE, there are no significant differences between the mean and median behavior. In contrast, LURE occasionally does suffer from high variance (mean behavior), e.g. for CIFAR-10 in Figure C.3. As discussed in the main body of the paper, this happens when the weight assigned by the proposal diverges strongly from the observed outcome for individual observations. This does not happen for ASE because, unlike LURE, we do not weight observations against predictions from the surrogate.

We also include versions of the main figures of the paper that normalize the squared errors of the estimators by the squared true test loss of the main model $f$ on the full test set. See Figure C.5, Figure C.6, and Figure C.7 for these plots. For the experiments in §7.1 and §7.2, the cross-entropy loss of the main model is $1.51 \pm 0.18$ (average over 100 runs) and the accuracy is $(85.63 \pm 0.25)\%$. For the experiments of §7.3, the cross-entropy losses are $0.961 \pm 0.039$ for CIFAR-100, $0.289 \pm 0.021$ for CIFAR-10, and $0.172 \pm 0.015$ for Fashion-MNIST. The accuracies are $72.54\%$ for CIFAR-100, $90.92\%$ for CIFAR-10, and $92.05\%$ for Fashion-MNIST. (We provide standard deviations on accuracies for the experiments of §B.4.)

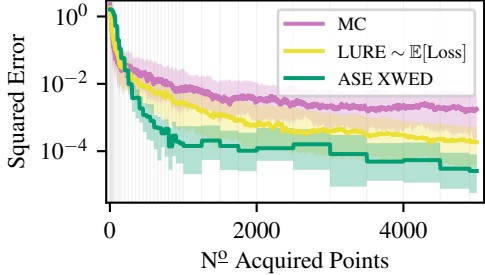

Figure C.1: Version of Figure 1 with medians and quantiles instead of means and standard errors. Convergence of the squared error for the distribution shift experiment. Shown are median squared errors and the shading indicates the 25/75 % quantiles over 100 runs. We see that ASE provides significant improvements over both LURE and MC. The vertical grey lines mark where we retrain the secondary model.

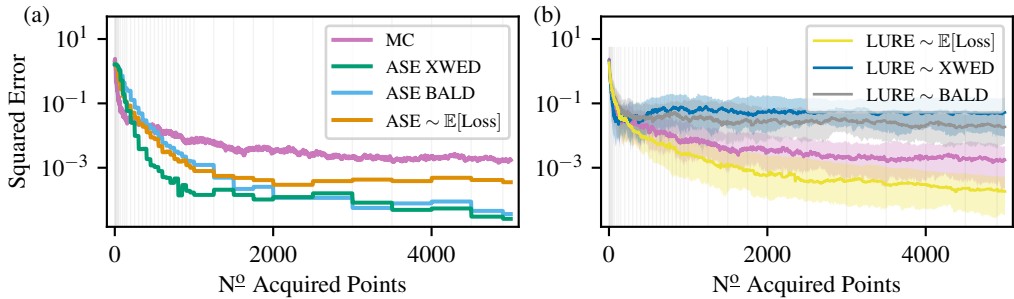

Figure C.2: Version of Figure 3 with median squared errors. (a) While XWED performs best, ASE outperforms MC for all investigated choices of acquisition functions. Shown are median squared errors over 100 runs, and we omit quantile shading to aid legibility. (b) LURE suffers high error when not acquiring from expected loss and instead using a different acquisition strategy. Shown are median squared errors and the shading indicates 25/75 % quantiles over 100 runs. We retrain $\pi$ at steps marked with grey lines. The '$\sim$'-symbol indicates stochastic acquisition.

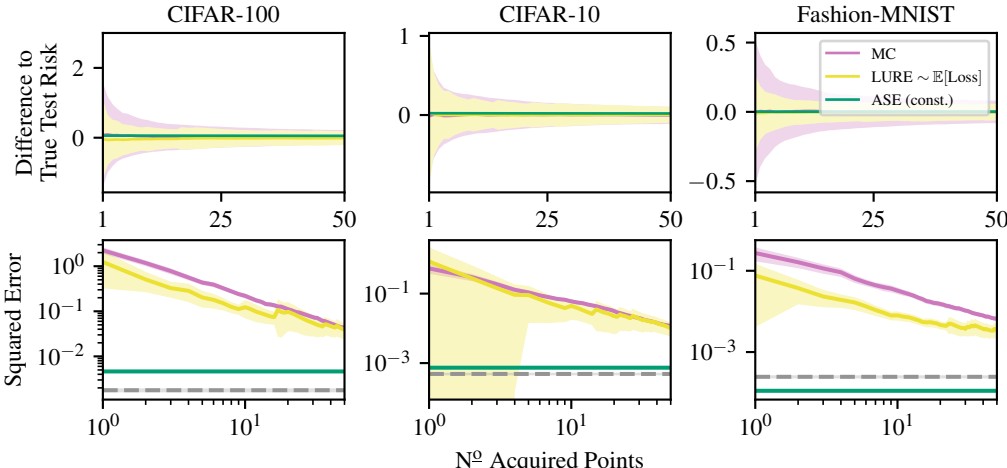

Figure C.3: Version of Figure 4 with means and standard errors. We display means over 1000 random test sets and shading indicates standard deviations for the differences (top, too small to be visible for ASE) and two standard errors for the squared error (bottom, too small to be visible for ASE and naive MC). Active Testing of ResNets on CIFAR-100, CIFAR-10, and FashionMNIST without retraining the auxiliary model $\pi$. Despite not observing any test labels, ASE again improves upon LURE. The grey line indicates the lower bound on the error from the finite size of the pool.

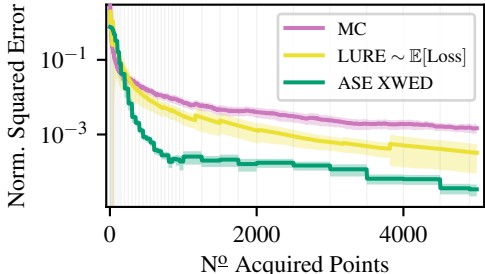

Figure C.5: Version of Figure 1 with squared error of the estimates normalized by the true value of the test error.

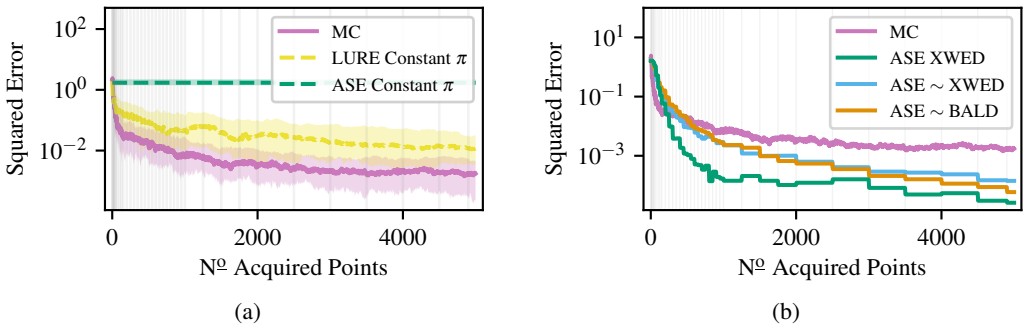

(a)

(b)

Figure C.4: (a) Version of Figure B.1 (a) with medians and quantiles instead of means and standard errors. Convergence of the squared error for the distribution shift experiment with constant secondary model $\pi$. Shown are median squared errors and the shading indicates the 25/75 % quantiles over 100 runs. Again, we see naive MC is not outperformed if $\pi$ is constant. (b) Version of Figure B.1 (b) with medians instead of means. Comparison of *stochastic* acquisition strategies for ASE on the distribution shift experiment. Shown are median squared errors over 100 runs and we omit quantiles for reasons of legibility. The vertical grey lines mark where we retrain the secondary model.

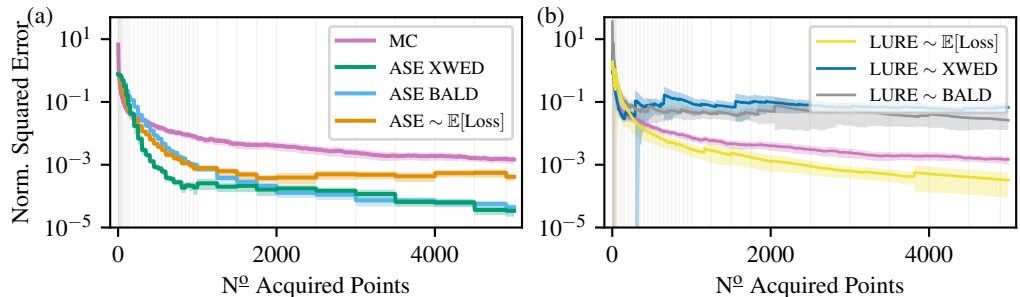

Figure C.6: Version of Figure 3 with squared error of the estimates normalized by the true value of the test error.

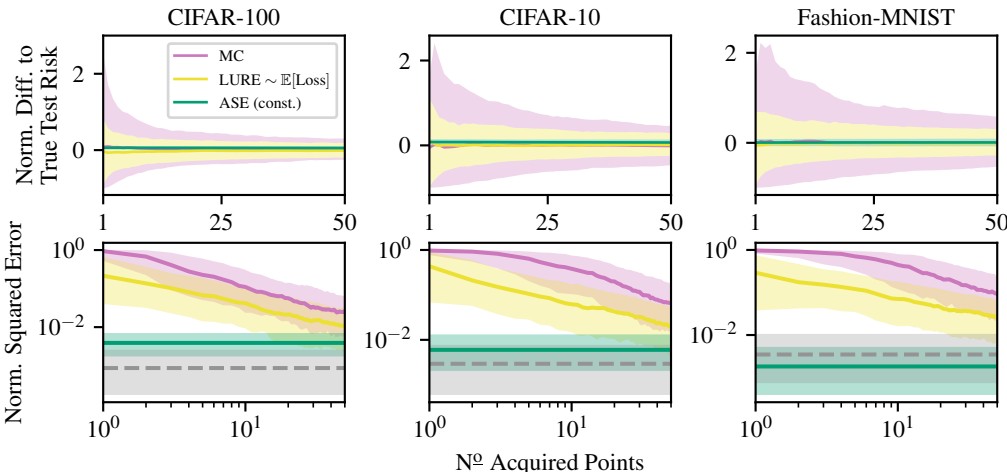

Figure C.7: Version of Figure 4 with differences and squared errors of the estimates normalized by the true value of the test error.

# D Experimental Details

## D.1 Experiment Definition

In this section, we provide a more detailed explanation of how we combine different estimators and acquisition functions in the experimental evaluation in §7. For our purposes, there are two ingredients to a method for model evaluation: the estimator and the acquisition strategy.

In terms of estimators, this paper compares the naive MC estimator, $\hat{r}_{\text{iid}}$, introduced in §2, and the LURE estimator, $\hat{r}_{\text{LURE}}$, Eq. (2), against the ASE estimator, $\hat{r}_{\text{ASE}}$ Eq. (4).

For the MC estimator, the acquisition of labels from the remaining samples in the test pool is always uniformly at random, so in the figures in §7, we simply write "MC" to denote this naive baseline. For any other acquisition scheme, $\hat{r}_{\text{iid}}$ would no longer be unbiased. However, for ASE and LURE, we have the option to use a variety of acquisition functions $a$. In this paper, we consider XWED, Eq. (6), BALD, Eq. (5), and $\mathbb{E}[\text{Loss}]$, Eq. (3) as acquisition functions. For all of the these acquisition functions, at any iteration $m$, we evaluate the acquisition function on all remaining samples $\mathcal{U}$ in the test pool, obtaining a set of acquisition scores $\{a(x_i)\}_{i \in \mathcal{U}}$.

Given these scores, there are two ways in which we can determine the next acquisition: deterministic and stochastic acquisition. In deterministic acquisition, we acquire a label for the sample $x_{i_m}$ remaining in the test pool with the largest acquisition score $i_m = \arg\max_i a(x_i)$. For stochastic acquisition, we choose the next point to acquire by sampling from the distribution over acquisition scores: we define a probability mass function over the indices as $p(i) = a(x_i)/\sum_{i \in \mathcal{U}} a_i$ and then sample an index $i_m \sim p(i)$ for acquisition. For the LURE estimator, acquisition needs to be stochastic for the importance sampling estimator to remain unbiased, while ASEs are compatible with both stochastic and deterministic acquisition.

In the experiments in §7, we add a "$\sim$" prefix for experiments with stochastic acquisition, and we add no prefix for deterministic acquisition. For example, "LURE $\sim$ BALD" is an experiment that uses the LURE estimator and stochastically samples acquisitions from the BALD scores, while "ASE XWED" denotes a run that uses the ASE estimator and deterministically acquires labels for samples with the highest XWED score in each round.

## D.2 Computing XWED and Related Quantities

To give a practical example of how to compute some of the quantities related to the ASE and active testing, we now present a worked through example for the case in which the main model $f$ is a neural network and the surrogate $\pi$ a deep ensemble, both trained for a classification task.

Here, both $f(x)$ and $\pi(y|x)$ are classifiers whose outputs are vectors of dimensionality $C$, where $C$ denotes the number of classes. Each element $i$ of these vectors, represents the predicted probability that the observed class $y$ is equal to $i$. The quantity we wish to estimate is the average negative log-likelihood on the test set, also known as the cross-entropy loss of the test set. Our loss function is thus $\mathcal{L}(y, f(x)) = -\log f(x)_y$, i.e. the negative log probability the model $f$ assigns to the true class.

For classification tasks, we can evaluate expectations with respect to outcomes $Y$ exactly by enumerating all possibilities. Further, for the deep ensemble surrogate $\pi$, we model the posterior distribution of the deep ensemble surrogate parameters as $p(\theta) = \frac{1}{E} \sum_{e=1}^{E} \delta(\theta - \theta_e)$, where $\delta$ is the Dirac delta and $e \in \{1, \ldots, E\}$ enumerates the ensemble components. Together, this simplifies the XWED acquisition function to simple sums over class labels and ensemble components:

$$\text{XWED}(x) = \sum_{y=1}^{C} -\pi(y|x)\mathcal{L}(y, f(x))\log\pi(y|x) + \frac{1}{E}\sum_{e=1}^{E}\pi(y|x, \theta_e)\mathcal{L}(y, f(x))\log\pi(y|x, \theta_e)$$

where, lastly, we note that a posterior predictive distribution for deep ensembles can be obtained as the average prediction of the ensemble components,

$$\pi(y|x) = \int \pi(y|x, \theta)\pi(\theta)\,\mathrm{d}\theta = \frac{1}{E}\sum_{e=1}^{E}\int \pi(y|x, \theta)\delta(\theta - \theta_e)\,\mathrm{d}\theta = \frac{1}{E}\sum_{e=1}^{E}\pi(y|x, \theta_e).$$

### D.3 Training Setup

**Evaluation with Distribution Shift.** We follow Kossen et al. [38] in the setup of the radial BNN, using code and default hyperparameters provided by Farquhar et al. [18]: we employ a learning rate of $1 \times 10^{-4}$ and set the weight decay to $1 \times 10^{-4}$, in combination with the ADAM optimizer [35] and a batch size of 64. We use 8 variational samples during training and 100 variational samples during testing of the BNN. We use convolutional layers with 16 channels. We train for a maximum of 500 epochs with early stopping patience of 10, and we use validation sets of size 500. We further use the validation set to calibrate predictions via temperature scaling.

**Evaluation of CNNs for Classification.** We follow Kossen et al. [38], who follow DeVries & Taylor [14], for the setup of the ResNet-18 for the CIFAR-10 and Fashion-MNIST datasets. Concretely, we set the learning rate to 0.1, the weight decay $5 \times 10^{-4}$, and use momentum of 0.9 with an SGD optimizer and batch size of 128. We anneal the learning rate using a cosine schedule. We use early stopping on a validation set of size 5000, with patience of 20 epochs, and we set the maximum number of epochs to 160. We use the validation set to calibrate predictions via temperature scaling.

Similarly, we follow Kossen et al. [38] for the setup of the WideResNet on CIFAR-100: we set the depth to 40 and again follow DeVries & Taylor [14] to set hyperparameters. Specifically, we train for 200 epochs, setting the learning rate to 0.1, the weight decay to $5 \times 10^{-4}$, and the momentum to 0.9. We use the SGD optimizer with a batch size of 128. We decrease the initial learning rate by a factor of 0.2 after 60, 120, and 160 epochs. Again, we use the validation set to calibrate predictions via temperature scaling.

## E   Computational Complexity

The computational complexity of ASEs can be split into two components: the computational complexity of the ASE estimate and the computational complexity of the acquisition process.

The ASE estimate requires us to predict with the surrogate model on each of the $N$ points in the test pool, cf. Eq. (4), leading to a computational complexity of $\mathcal{O}(N)$.

During acquisition, at each iteration $m$, we need evaluate the acquisition function on all $N - m$ remaining samples in the pool. This is repeated for all $M$ iterations leading to an overall computational complexity of $\mathcal{O}(MN)$. Additionally, we have to consider the computational complexity of retraining the surrogate model between iterations. We assume, for worst case complexity, that the surrogate is retrained in each of the $M$ iterations. Further, we assume that the complexity of training the surrogate is proportional to the size of the dataset. If we assume an initial training set of size $|\mathcal{D}_{\text{train}}|$ for the surrogate, then in each testing iteration $m$, we incur additional computational complexity of $\mathcal{O}(m + |\mathcal{D}_{\text{train}}|)$, or $\mathcal{O}(M \cdot (M + |\mathcal{D}_{\text{train}}|))$ over all iterations.

In total, this gives the computational complexity of ASEs as $\mathcal{O}\left(N + M \cdot (N + M + |\mathcal{D}_{\text{train}}|)\right)$. Given that $M \leq N$ always, this can be simplified to $\mathcal{O}\left(M \cdot (N + |\mathcal{D}_{\text{train}}|)\right)$.

Similarly, Kossen et al. [38] derive the computational complexity of active testing with LURE as $\mathcal{O}(M \cdot (N + |\mathcal{D}_{\text{train}}|))$. Therefore both ASE- and LURE-based active testing have identical big-$\mathcal{O}$ complexity. Trivially, the naive Monte Carlo baseline simply has a much lower complexity of $\mathcal{O}(M)$ than either active testing approaches. We note that, while naive Monte Carlo is cheaper for given number of acquisitions $M$, the expense of ASEs and LURE is justified in the active testing scenario by the assumption that these labels are expensive to acquire. Note here that the *labelling* cost of ASE and LURE is exactly the same as MC (and thus $\mathcal{O}(M)$).

## F   Societal Impact

With ASEs, we present a general framework for the evaluation of machine learning models. It is therefore hard to evaluate the societal impact of ASEs without restricting the discussion to one of many possible areas of application. As with any application of machine learning in critical contexts, we advocate that the alignment of model predictions and societal values is thoroughly evaluated. Below, we present some more specific thoughts we hope may be useful when considering the societal impact of ASEs in a specific application.

**Biases in Surrogate Predictions.** Estimates from ASE are inherently subjective as they rely directly on the predictions of the surrogate model. Because of this, decisions based on estimates from ASE should be more carefully monitored than those based on MC. In particular, one usually would only have to consider potential unfairness/societal bias in the original model, but for ASE these also need to be considered as part of the estimation itself.

**The Definition of 'Expensive'.** ASEs assume that label acquisition is more expensive than the computational cost required to iteratively train and retrain the surrogate model. Depending on the application, the goals of efficient evaluation and the goals of society may or may not be aligned. We could, for example, assume that the ASE user cares about saving money, while society cares about the $CO_2$ footprint of the approach. In some applications, labels might be really expensive in terms of their monetary price but might not require much $CO_2$, e.g. expert annotations in medical applications. In this case, the goals of the ASE user and society might not be aligned because society would prefer more expert queries rather than $CO_2$-intensive retraining of large surrogate models. On the other hand, labels might be expensive *and* $CO_2$-intensive, e.g. when the function is a computationally demanding numerical simulation. In this case, the goals of society and the ASE user may be aligned because being efficient in labels benefits both. Note that this discussion similarly applies to both active testing and active learning more generally.

# G    Compute Details, Licenses, and Resource Citations

**Compute Details.**    All experiments were run on desktop machines in an internal cluster with either a NVIDIA GeForce RTX 2080 Ti (12 GB memory) or a NVIDIA Titan RTX (24 GB memory) graphics card.

A single run for the experiment of §7.1 and §7.2 takes about 8 hours on a single machine. For the §7.3, the experiments with the Resnet-18 ensemble on Fashion-MNIST take about 9 hours to compute, and 14 hours for the Resnet-18 ensemble on CIFAR-10. The results for the WideResNet ensemble on CIFAR-100 can be computed in 24 hours, if ensemble members of $\pi$ are trained in parallel across different GPUs.

**License Agreements and Resource Citations.**
*License agreement of the MNIST dataset [42]*: License: Yann LeCun and Corinna Cortes hold the copyright of MNIST dataset, which is a derivative work from original NIST datasets. MNIST dataset is made available under the terms of the Creative Commons Attribution-Share Alike 3.0 license.

*License agreement of the Fashion-MNIST dataset [69]*: Fashion-Mnist published under MIT license.

*License agreement of the CIFAR-10 and CIFAR-100 dataset [39]*: CIFAR-10 and CIFAR-100 are published under MIT license.

*License agreement of Python [66]*: Python is dual licensed under the PSF License Agreement and the Zero-Clause BSD license.

*License agreement of PyTorch [52]*: PyTorch is licensed under a modified BSD license, see `https://github.com/pytorch/pytorch/blob/master/LICENSE`.