# OpenReview forum: "Active Surrogate Estimators: An Active Learning Approach to Label-Efficient Model Evaluation"
_NeurIPS.cc/2022/Conference — NeurIPS 2022 Accept_

### Official Review · Reviewer_DGAC · 2022-07-11

**Rating:** 8
**Confidence:** 4
**Soundness:** 4 excellent
**Presentation:** 3 good
**Contribution:** 4 excellent

**Summary:**

The authors proposed a new method, ASE, for label-efficient model evaluation. ASE uses a surrogate-based estimation approach that interpolates the errors of points with unknown labels. ASEs actively learn the surrogate. The authors proposed a novel acquisition strategy, XWED, that tailors this learning to the final estimation task. Experiments show that ASEs offer greater label-efficiency than SOTA.


**Questions:**

1) Could we use the difference between expected accuracy and current accuracy as a metric, besides squared error, as the metric to evaluate different algorithms? If so, how is the performance compared to other active learning algorithms, e.g., BALD and Entropy?
2) Could we evaluate these algorithms in a batch mode setting? For each algorithm, we can specify the batch size $B$ and select top-$B$ in each round. We believe it is very interesting to extend XWED to batch mode setting, similar to the way in which BatchBALD extends BALD.

**Limitations:**

could the authors also discuss the computation complexity in the paper for different algorithms?

**Strengths And Weaknesses:**

1) The authors proposed a novel algorithm based on surrogate estimation for active testing.
2) The authors maintain a high quality of writing and presentation.

---

> ### Author Response · Authors · 2022-08-02
> **Author Response to Reviewer DGAC**
>
> Thank you for your hard work and helpful feedback! We are delighted by your positive endorsement of our work.
>
> > Could we use the difference between expected accuracy and current accuracy as a metric, besides squared error, as the metric to evaluate different algorithms?
>
> Yes, absolutely!  ASEs can be used with arbitrary loss functions $\mathcal{L}$, and for the 0-1 loss function, $\mathcal{L}(f(x), y)= \mathbb{1}[f(x) = y]$, ASEs can be used to estimate test accuracies directly. Following your excellent suggestion, __we have added a new experiment in Appendix B.4 that studies ASEs and baselines for accuracy estimation__. The experiment shows that ASEs also give SOTA performance for this problem.
>
> > Could we evaluate these algorithms in a batch mode setting? For each algorithm, we can specify the batch size B and select top-B in each round. We believe it is very interesting to extend XWED to batch mode setting, similar to the way in which BatchBALD extends BALD.
>
> This is an excellent suggestion! Batch acquisitions with ASEs should indeed be achievable: the ASE estimator itself works equally well whether samples are acquired individually or in batches, while the XWED acquisition strategy can easily be extended in exactly the way you describe, noting that we can weight by the loss in the BatchBALD acquisition function in exactly the same way we currently adjust BALD.  We will add discussion on this to the final version of the paper and look to add evaluations if possible as well.   We also note here that batch settings offer further potential advantages of ASEs over LURE, as the latter cannot be straightforwardly extended to the batch setting in the same way.
>
>
> > Could the authors also discuss the computation complexity in the paper for different algorithms?
>
> Thank you for suggesting this. __We have added a discussion of the computational complexity to the supplementary material in Section E, "Computational Complexity".__  In short, the computational complexities of ASEs and LURE are identical, while standard Monte Carlo is naturally much cheaper for a given number of labels (with the expense of ASEs/LURE justified by the assumption that these labels are difficult and/or expensive to acquire).
>
> Again, thank you for the review. Please let us know if you have any further questions or suggestions, or if you would like any additional clarifications.

---

### Official Review · Reviewer_qkQm · 2022-07-11

**Rating:** 6
**Confidence:** 4
**Soundness:** 3 good
**Presentation:** 3 good
**Contribution:** 3 good

**Summary:**

This paper proposes the ASE, showing a better performance than the Lure approach in active testing. The newly proposed networks for the g and \pi function and the acquisition function show better performance in the active test. Also, the various aspects of ASE are explored in the distribution shift.


**Questions:**

I have some questions conserving the active test and algorithms.
1. The approximation of risk or loss can be valuable. However, the minimum loss cannot ensure maximum accuracy in the classification task. In general, active learning and accuracy is essential. Let me know your answer.
2. What's the primary reason for reducing the variance of active sampling in the AES. Updating procedures or architectures. Can you provide more solid answers?
3. Does the network's capacity for \pi influence the performance of AES?



**Ethics Review Area:**

["I don’t know"]

**Limitations:**

There is no critical flaw in this paper. The section of theoretical aspects can be improved in the presentation. Experiments are limited, related the selection of \pi.

**Strengths And Weaknesses:**

Strong points: The proposed algorithm is intuitive and straightforward, performs better, and addresses the active test problem well. Experiments are well designed to show the critical properties of the proposed algorithm.

Weak points: Theoretical analysis of the ASE error is somewhat vague. The decompositions and analysis are valuable. However, I, II, and III properties are not well-explained, including too massive contexts. It seems that the main is the empirical analysis due to the non-existence of any theorem,

---

> ### Author Response · Authors · 2022-08-02
> **Author Response to Reviewer qkQm**
>
> Thank you for your hard work and helpful feedback!
>
> > Theoretical analysis of the ASE error is somewhat vague. The decompositions and analysis are valuable. However, I, II, and III properties are not well-explained, including too massive contexts.
>
> We are happy you value the decomposition of the ASE error. We wish to emphasize that the 'vagueness' of the theoretical result is a consequence of its generality and the infeasibility of providing more exact analysis without assuming a specific form of surrogate (along with additional assumptions about the true loss). Nonetheless, __we have revisited the presentation of the individual terms to clarify their exposition, including providing a new intermediary breakdown of the terms__. Thanks for bringing this to our attention.
>
> > The approximation of risk or loss can be valuable. However, the minimum loss cannot ensure maximum accuracy in the classification task. In general, active learning and accuracy is essential.
>
> Thanks for bringing this up. We agree that accuracy is an important metric for model evaluation. ASEs are compatible with arbitrary loss functions. Thus, by using the 0-1 loss function, ASEs can be used to target model test accuracies directly.  Following your suggestion, __we have added a new experiment in Appendix B.4 that studies ASEs and baselines for accuracy estimation__, finding that ASEs again give SOTA performance.
>
> > What's the primary reason for reducing the variance of active sampling in the AES. Updating procedures or architectures.
>
> Our architectures are shared with those used by the LURE baseline and so the lower variance estimates are from the methodological advancements of ASEs, rather than using more powerful networks or any low-level tuning.  As shown in Figure 2, both the use of the general ASE framework and our XWED acquisition strategy are important components for achieving these gains.  Reducing the variance provides more accurate model evaluation, which is, in turn, useful for a wide variety of tasks, such as guiding model/architecture construction, hyperparameter tuning, and empirical verification of model performance.
>
> > Does the network's capacity for $\pi$ influence the performance of AES?
>
> Yes, this certainly can be a consideration: it is important that $\pi$ is chosen from a class of models that is sufficiently powerful. Term (IIA) (the old Term (II)) in our theoretical analysis (Eq. 7) represents exactly the error that is introduced to the final estimate from using a $\pi$ with insufficient capacity to model the true conditional process $Y|x$. However, note that this will rarely be a concern in practice, as typically the error from the finite number of labels available in training $\pi$ (i.e. Term (IIB)) will dominate the error from the theoretical network capacity.  __We have made edits to further emphasize the importance of $\pi$ to ASE’s performance.__
>
> Again, thank you for the review. Please let us know if you have any further questions or suggestions, or if you would like any additional clarifications.

---

> > ### Comment · Reviewer_qkQm · 2022-08-07
> > **Reply to Authors’ response**
> >
> > Thanks for your reply.
> > Many issues are well-addressed.
> > Maybe in theoretical aspects, there can be many issues. However, this paper concerns the algorithm.
> > Also, I believe that the architecture well-cooperated with $\pi$ and the proposed acquisition function can be studied, which can be a valuable future work.

---

### Official Review · Reviewer_Aog1 · 2022-07-15

**Rating:** 6
**Confidence:** 4
**Soundness:** 3 good
**Presentation:** 3 good
**Contribution:** 3 good

**Summary:**

This paper proposes a new method for active testing called Active Surrogate Estimators (ASEs). ASEs actively learn a surrogate model to predict the losses and also employ a novel acquisition strategy, called XWED, which is a modification of BALD but puts weights for each data point using its corresponding loss value. The experiments show the proposed method is better than other baselines such as MC and LURE.

**Questions:**

In Eq (6), why don't we just multiply a factor $\mathbb{E}_{Y \sim \pi(\cdot|x)} [\mathcal{L}(Y, f(x))]$ directly into $\mathrm{BALD}(x)$ in Eq (5) ? Is there any specific reason why we need to multiply $\mathcal{L}(Y, f(x))$ into each of the expectations in Eq (6) ?

**Limitations:**

No potential negative societal impact.

**Strengths And Weaknesses:**

Strengths:
- The problem considered is important.
- The proposed methods, ASEs and XWED, are novel and well-motivated.
- The proposed methods achieve good performance in the experiments.
- The paper is well-written, although the presentation can be further improved.

Weaknesses:
- The proposed methods are small modifications of previous methods. Particularly, ASE is a small modification of LURE, while XWED is a small modification of BALD. However, I think the proposed methods are still good contributions since these modifications help improve the performance of the methods in practice.
- The paper does not give adequate descriptions for several baselines used in the experiments, e.g. LURE ~ $\mathbb{E}$[Loss], LURE ~ XWED, LURE ~ BALD, ASE BALD, etc. It would be better to explain each of these methods using some formulas. That would help readers understand more about the ways different components of the methods can be used.
- I think it would also be better for the paper to give some concrete examples where the proposed methods are applied and computed. For example, what would be the specific formulas when the model is a neural network, what would be the model $\pi$, and how the computation is done? These examples can be given in reference to specific steps in Algorithm 1. I think Section 5 of the paper can be shorten for more space.
- There is also a related research area that aims to evaluate models without any labels, e.g. see [1, 2] below and other related work. It would be better if the paper also compares their active testing algorithm with these methods.

References:
- [1] Corneanu et al. Computing the Testing Error without a Testing Set. CVPR 2020.
- [2] Deng & Zheng. Are Labels Always Necessary for Classifier Accuracy Evaluation? CVPR 2021.

---

> ### Author Response · Authors · 2022-08-02
> **Author Response to Reviewer Aog1**
>
> Thank you for your hard work and helpful feedback! We have very gladly incorporated your excellent suggestions and hope the following responses alleviate any remaining concerns you might have.
>
> > The paper does not give adequate descriptions for several baselines used in the experiments, e.g. LURE ~ E[Loss], LURE ~ XWED, LURE ~ BALD, ASE BALD, etc. It would be better to explain each of these methods using some formulas.
>
> Thank you for this excellent suggestion, we agree that the paper would benefit from clearer and more comprehensive explanations of these. To this end, __we have added Section D.1, "Experiment Definition"__, in the supplement to include a larger background section on how exactly the individual experiments for ASE and baselines are constructed. For the final version of the paper (which allows 10 pages rather than the current 9), we plan to promote the bulk of these descriptions to the main paper.
>
> > I think it would also be better for the paper to give some concrete examples where the proposed methods are applied and computed. For example, what would be the specific formulas when the model is a neural network, what would be the model π, and how the computation is done?
>
> Again, thank you for this great suggestion. __We have updated the supplement to include a more in-depth description of how the terms are computed for the exact setting of our experiments, see Section D.2.__ We plan to include this discussion in the main paper for the camera ready version, as we agree this is important information for practitioners.
>
> > There is also a related research area that aims to evaluate models without any labels, e.g. see [1, 2] below and other related work. It would be better if the paper also compares their active testing algorithm with these methods.
>
> Thank you for pointing out these papers. We agree that they are relevant and __have added a discussion of them and other additional related work in Appendix F, “Additional Related Work”__. For the final version of the paper, we will move this discussion to the main paper.
>
> Unfortunately though, we do not think direct empirical comparisons are feasible, because [1] and [2] do not target quite the same problem.  Namely, at a high-level, both papers are based around performing meta analysis and require access to multiple related datasets; they rely on regressions from dataset/model summaries to performance scores.  Neither can be used with access to only a single dataset and so cannot be applied in the setting we consider.  Further, unlike ASEs, they assume that there is no access to any test labels at all and they do not themselves provide comparison to conventional estimators utilizing test labels like we do.
>
> Though these differences in problem settings mean they are not directly competing methods, it is still interesting to note that the errors of our ASE estimators tend to be much lower than those in [1] and [2]. Namely, [1] claims that ‘an average error between 8.45% and 4.60% is obtained across computer vision problems’,  while the average of the error values reported in [2] is 2.4%. In contrast, ASEs obtain an average error of 0.91% (CIFAR-100), 0.45% (CIFAR-10), and 0.74% (Fashion-MNIST) for estimating accuracy (see our new Appendix B.4).
>
> > In Eq (6), why don't we just multiply a factor  $\mathbb{E}_{Y∼π(⋅|x)}[\mathcal{L}(Y,f(x))]$  directly into  $\mathrm{BALD}(x)$  in Eq (5) ? Is there any specific reason why we need to multiply  $\mathcal{L}(Y,f(x))$ into each of the expectations in Eq (6) ?
>
> Good question! Though acquiring from $\mathbb{E}_Y[\mathcal{L}(Y,f(x))] \cdot \mathrm{BALD}(x)$ would not be crazy, we believe there are a number of reasons to prefer XWED:
> - Unlike factorizing by the expected loss, XWED can still be viewed as conforming to a formal Bayesian experimental design decision framework, in that it is equal to the expectation of a utility over experiment outcomes.  Namely, it corresponds to using the utility function $u(Y,x) = \mathbb{E}_{\Theta\sim\pi(\cdot|Y,x)}[\mathcal{L}(Y,f(x))(\log \pi(Y|x,\Theta)-\log \pi(Y|x))]$.
> - XWED will account for interactions between disagreement and loss as a function of $Y$ because it takes a (more flexible) expectation of products, rather than a product of expectations.  For example, one could have a point with both high expected loss and high expected disagreement, but where there are no possible outcomes that yield a high loss and high disagreement at the same time (such that $u(Y,x)$ is itself never large).  Here XWED will be small, but the product of expectations will be large.
> - We tend to find XWED performs slightly better empirically.
>
> We will add details on this to the final version of the paper to strengthen the justification for XWED.
>
> Again, thank you very much for your review; we believe your suggestions have noticeably helped strengthen the paper. Please let us know if there are any further changes you would like to see or if there is anything else that we can clarify.

---

> > ### Comment · Reviewer_Aog1 · 2022-08-08
> > **Reply to Author Response**
> >
> > Thanks for the clarifications.

---

> > > ### Author Response · Authors · 2022-08-08
> > > **Author Response to Reviewer Aog1**
> > >
> > > Thanks for taking the time to read our rebuttal.

---

### Official Review · Reviewer_mCJr · 2022-07-15

**Rating:** 6
**Confidence:** 4
**Soundness:** 3 good
**Presentation:** 4 excellent
**Contribution:** 3 good

**Summary:**

Extensive research is going on in an active learning area. However, the annotation cost of an evaluation set is overlooked in previous studies. Since active learning assumes expensive labeling costs, the cost of evaluation set annotation is also important. Active testing, the problem addressed in this paper, has been underexplored.

This work tackles how to minimize a labeling cost for an evaluation set. The paper proposes two main ideas: an estimator function that predicts the expected error of a task model, and an acquisition function to minimize the proposed expected error function. The paper experiments with its method on MNIST dataset with distribution shift scenario, and error estimation without distribution scenario on CIFAR10, CIFAR100, and F-MNIST.

**Questions:**

1. How is R_ASE sensitive to the accuracy of a surrogate model?
2. What is the absolute scale of true losses in the experiments? How would the graph look like if graphs are scaled (normalized) over true loss?


**Limitations:**

The supplementary material addresses societal impact and limitations.
However, the most important limitation, the dependency of R_ASE (equation.4) and surrogate model accuracy not analyzed nor addressed in this paper.


**Strengths And Weaknesses:**

## Strength

### Novelty
The paper proposes a new error estimation function (Active surrogate estimator) and test-data acquisition function.

### Experiment
The paper shows its performance with and without a distribution shift scenario, addressing the effect of the method on a scenario with a missing class scenario (an extreme case of class imbalance).
Section 7.2 shows the performance on CIFAR10, CIFAR100, and F-MNIST which are sufficient to show their effectiveness in terms of experiment size.
In both with/without distribution shifts, the proposed method is meaningfully effective over the competing methods.

### Analysis
Section5 decomposes the error bound of an active surrogate estimator and interprets each term. The interpretation of these terms is meaningful and interesting.


## Weakness

### Weak novelty of surrogate estimator and its limitations
1. weak novelty: Although the paper proposes two parts, an active surrogate estimator and acquisition function of active testing, the surrogate estimator (Equation (4)) idea is not surprising. In my simplified interpretation, equation (4) is a loss prediction of a task model using soft-pseudo labels, i.e., the prediction of a surrogate model. In other words, this objective heavily relies on a surrogate model in hoping that the surrogate model is very accurate as true ground truths. In addition, most of the performance gain of this paper is coming from this surrogate estimator, not the acquisition function (figure2 (a))

2. Limitation: Due to the dependency of equation(4) on the surrogate model, I presume that the performance of the surrogate model is critical for equation (4) to perform well, which is an important limitation not analyzed nor addressed in this paper.

### Experiment with strong task and surrogate models
This experiment setting may be due to the limitation of equation (4).
In the experiment, I conjecture that the authors use strong task and surrogate models.

1. In section 7.1, the task and surrogate model are initially trained on 2000 data (roughly 3.5% of the MNIST training set. the Training set size of MNIST is 60,000). Typical classification models perform over 95% accuracy with 1% of the MNIST training set. This is not the weakness but I am pointing out that the task and surrogate model accuracy is high.

2. In section 7.2, experiments are conducted on more challenging datasets. In these experiments, both task and surrogate models are trained using 40,000 samples. This is 80% of the training dataset for CIFAR10 and CIFAR 100 which has a training set size of 50,000 samples.
In a typical active learning setting, models are evaluated starting from 10% and additionally trained on an actively annotated labeled set of size 20-30%, summing up a total of 30-40% annotated data. I presume the models are trained on a larger set of samples (40,000 / 80%) than the typical setting (18,000~2,4000 / 30-40% ) due to the dependency of equation (4) on surrogate model performance.

Missing analysis and limitation.
Since surrogate model performance important factor in this study, this should be analyzed and addressed in this paper.

### Missing details or experiment.
1. Section 7.1 shows only 1 case for the distribution shift analysis. It would be better to show a few more cases for this study.
2. The graphs in this paper show squared error, the difference between a true expected loss and predicted loss. What is the absolute size of the true expected loss? What would be the critical value for reasonable estimation error for loss expectation? What would the graph look like if you use the ratio between the expected loss and true loss?
3. Typo? In Equation (7), ‘R’ should be ‘r’?

---

> ### Author Response · Authors · 2022-08-02
> **Author Response to Reviewer mCJr**
>
> Thank you for your hard work and helpful feedback!  We hope that our responses below and the accompanying paper updates alleviate your concerns.  We would particularly like to draw your attention to our new experiment that shows ASEs still performing well with a weaker surrogate.
>
> > The surrogate estimator (Equation (4)) idea is not surprising [...] In addition, most of the performance gain of this paper is coming from this surrogate estimator, not the acquisition function (figure2 (a)).
>
> We would like to emphasize that the core idea of using a surrogate regression of the loss as basis for estimation is itself a core contribution of the paper.  Thus while Eq (4) follows directly from a combination of this idea and our suggested way of constructing the regressor, the innovation comes in the idea of using such an approach in the first place.
>
> Regarding the gains, ASEs are already a novel approach introduced by the paper regardless of the specific acquisition strategy used.  The baseline approaches in Figure 2 are Monte Carlo (MC) and LURE (with expected loss acquisition), with our suggested approach (ASE XWED) providing substantial gains over both.  Other acquisition strategies are included for ablation.  The only acquisition strategy previously suggested for the active testing context is expected loss, which actually performs relatively poorly for ASEs, with error comparable to LURE.  Thus, even though the differences between XWED and BALD are more modest—with substantial gains before around 1000 steps of acquisition but not after—both the switch to using ASEs and a deviating from previous active _testing_ acquisition is important.
>
> >  I presume that the performance of the surrogate model is critical for equation (4) to perform well, which is an important limitation not analyzed nor addressed in this paper. [...]
>
> > Q1. How is R_ASE sensitive to the accuracy of a surrogate model?
>
> You are indeed correct that the performance of the surrogate model is important to the performance of the ASE. In fact, there are only two sources of error in the ASE: error from the finiteness of the test pool and error from the surrogate. This is exactly the focus of the theoretical analysis in Section 5, and __we have updated the presentation in Section 5 in the revised version of the paper to make this clearer__, introducing an intermediary breakdown of the error into exactly these terms, before breaking down the latter into the old Terms (II) and (III).  In Section 5.1 we discuss and empirically quantify the size of these contributions to the error of the ASE, concluding that Term (IIB) (the old Term (III)) will typically dominate in practice.
>
> Thus to directly answer Q1, the performance of $r_\text{ASE}$ is typically exactly equivalent to the performance of the surrogate model, remembering though that the latter is not generally fixed as we can actively learn the surrogate as we acquire more labels.  From this, we can conclude that ASEs will be beneficial when we have access to / can actively learn surrogates with lower error than Monte Carlo based estimation (akin to the trade-off between variational and Monte Carlo methods); our experiments provide specific examples of where this is the case.  Note here that we are not trying to claim that ASEs will always be beneficial to MC estimation, but model evaluation is an extremely broad and common problem, and there are still lots of cases where we expect to be able to learn effective surrogates and leverage the benefits of ASEs.
>
> In light of the above, we disagree with the suggestion that there is no analysis or addressing of this “most important limitation;” we believe that we provide both theoretical and empirical analysis of it. However, we do agree that this could have been made clearer and the potential limitations more explicitly discussed. __We have therefore made edits to highlight this in Section 5__, while we plan to add further discussion on this to Section 8 in the final version of the paper.
>
> **Please see the next message for part two of our reply.**

---

> > ### Author Response · Authors · 2022-08-02
> > **Author Response to Reviewer mCJr (Part Two)**
> >
> > (This is part two of our reply to reviewer mCJr. Please read part one first.)
> >
> > > In the experiment, I conjecture that the authors use strong task and surrogate models. [...]
> >
> > > Typical classification models perform over 95% accuracy with 1% of the MNIST training set
> >
> > While we agree that ASEs benefit from strong surrogate models (see discussion above), we would like to clarify that they do not require the main task model to be strong; the task model being weaker does not generally make it harder to evaluate.  In fact, it will typically be easier to evaluate such models as their mistakes are more straightforward for the surrogate to spot.
> >
> > We would also argue that the surrogate (and indeed task) model used in Section 7.1 is actually not especially “strong.”  Even though the training dataset contains 2000 points, it contains no 7s at all and thus starts with poor accuracy before any test data is observed.  This experiment thus shows how it is possible to learn an effective surrogate on the fly, rather than requiring one upfront.
> >
> > > I presume the [CIFAR] models are trained on a larger set of samples (40,000 / 80%) than the typical [active learning] setting (18,000~2,4000 / 30-40% ) due to the dependency of equation (4) on surrogate model performance.
> >
> > The training data we are using here is not itself actively chosen and so we were not trying to replicate the typical setups for actively learned models, noting that active testing can be used independently of whether the original model was actively learned or not.  The reason for the slightly unusual choice of training on 80% of the training data was actually based on trying to balance being close to conventional, non-active, CIFAR training (where one would usually use the full training dataset) and maintaining a large as possible evaluation set to ensure we can accurately evaluate the performance of the different estimators, noting that the ASE error is approaching that of the full pool estimate.
> >
> > Nonetheless, we felt that it was a great idea to investigate how the performance of ASEs varies with the size of the training dataset for this experiment, as this can provide an ablation of how performance varies with surrogate strength. To this end, __we have added a variation of this experiment to Appendix B.5, where we restrict the size of the training set to only 10,000 points__, less than would usually be used in a typical active learning setting, thus yielding a relatively weak surrogate model. We find that ASEs continue to outperform all baselines in this challenging scenario, with very similar relative performances to those already quoted using the larger dataset.
> >
> > Again, thanks for this excellent suggestion: we believe the additional results underline the robustness of our approach, while we plan to extend this to a full set of ablations over training set sizes for the final version of the paper.
> >
> > > Section 7.1 shows only 1 case for the distribution shift analysis. It would be better to show a few more cases for this study.
> >
> > We agree that additional experiments of this case would be helpful and will look to add them in for the final version of the paper.
> >
> > > What is the absolute scale of true losses in the experiments? How would the graph look like if graphs are scaled (normalized) over true loss?
> >
> > Good question.  In short, the plots look very similar as the true risks are all between $0.17$ and $1.51$, while the fact that the true risk is not a function of the active testing method means the scaling just becomes a constant offset given that the y-axis is log scaled.  We have updated the supplementary material to include versions of all figures of the main paper which display squared errors and differences normalized by the true loss of the model. __Please see the last paragraph in Appendix C, "Additional Figures," where we have also further added the statistics of the 'true' losses and accuracies on the full test set.__
> >
> > > Typo? In Equation (7), ‘R’ should be ‘r’?
> >
> > Thank you, you are correct! This is a typo which we have now fixed.
> >
> > Again, thank you very much for your review. We hope that our reply has addressed your concerns. Please let us know if there are any further changes you would like to see or if there is anything else that we can clarify.

---

> > > ### Comment · Reviewer_mCJr · 2022-08-03
> > > **Thank you for your detailed feedback.**
> > >
> > > Thank you for your detailed feedback and additional experiments.
> > >
> > > - I like how the authors revised section 5.
> > > - The author's feedback to my question about 'the use of strong surrogate models' and additional experiments shown in Appendix B.5 are convincing.
> > >
> > >
> > > All of my important concerns are well addressed in the author's feedback.
> > > I appreciate the authors' effort and I would gladly increase my rating.

---

> > > > ### Author Response · Authors · 2022-08-04
> > > > **Author Response to Reviewer mCJr**
> > > >
> > > > Thank you for your careful evaluation of our rebuttal! We are delighted that you feel our reply addresses your concerns well. Do not hesitate to reach out if anything else should come up that we can help clarify.

---

### Author Response · Authors · 2022-08-02
**Author Response to All Reviewers**

We thank all the reviewers for their careful consideration, insightful comments, and helpful suggestions. We are glad that the paper was generally well received, and hope that our responses and paper updates alleviate the concerns that were raised.  We believe your input has already helped improve the paper and look forward to engaging with you further during the discussion period.

We are pleased that ASEs (and the XWED acquisition function) were received as 'new' (mCJr, Aog1, qkQm, DGAC), 'novel' (DGAC, Aog1), and 'well-motivated' (Aog1) contributions that 'address the active testing problem well' (qkQm).
We are grateful for the praise of our empirical results as 'well designed to show the critical properties of the proposed algorithm' (qkQm), and demonstrating that ASEs are 'meaningfully effective' (mCJr), 'offer greater label-efficiency than SOTA' (DGAC), and 'achieve good performance [...] better than other baselines' (Aog1). We also appreciate the highlighting of our submission as 'maintain[ing] a high quality of writing and presentation' (DGAC), being 'well-written' (Aog1), and reviewer mCJr rating the quality of the presentation as 'excellent'.

Following your excellent feedback, we have added two new experiments:

1. In Appendix B.4, we apply ASEs to the estimation of _model accuracy_ instead of cross-entropy loss, with SOTA performance again obtained.
2. In Appendix B.5, we provide an ablation of the experiments in Section 7.3 in which we decrease training set size to 10,000 samples. This reduces the quality of the surrogate model and thus makes the problem more challenging for ASEs. Despite this, we find that ASEs maintain their significant advantage over the baseline approaches.


We address your specific comments in the individual replies below.  We note that because the page limit for the revision is still 9 pages, while a 10th will be allowed for the camera ready, many of our paper updates and added content (including the experiments above) are currently in the supplementary material rather than the main paper.  However, we plan to move the bulk of them into the main paper for the final version.

---

### Meta-Review · Area_Chair_t6qk · 2022-08-28

**Recommendation:** Accept
**Confidence:** Certain

**Metareview:**

This paper tackles the active testing problem and proposes a novel estimator (ASE) of the expected loss based on a surrogate function, and a novel acquisition function (EWED) to minimize the error.  SOTA performance is reported, and some ablation study is conducted.

Reviewers raised concerns about novelty, high dependence on the quality of the surrogate, and missing baselines, which the authors have well addressed and convinced the reviewers.  Although the proposed methods are small modifications from existing methods, by using the surrogate, they significantly improve the existing methods in the experiments.

**Award:**

No

---

### Decision · Program_Chairs · 2022-09-14

Accept